

# Evolution of nonstationary hydrological drought characteristics in the UK under warming

Srinidhi Jha[1*], Lucy J. Barker[1], Jamie Hannaford[1,2], Maliko Tanguy[1,3]

[1]UK Centre for Ecology & Hydrology, Maclean Building, Crowmarsh Gifford, Oxfordshire, OX108BB, United Kingdom

[2]Irish Climate Analysis and Research UnitS (ICARUS), Maynooth University, Maynooth, Co. Kildare, Ireland

[3]European Centre for Medium Range Forecasts (ECMWF), Reading, United Kingdom

*Correspondence to: Srinidhi Jha (srijha@ceh.ac.uk)

**Abstract.** Although the United Kingdom (UK) is relatively wet, there is an increasing awareness of the impacts of droughts, and an expectation that droughts will become worse in the future. This has motivated studies that have developed projections of future UK drought characteristics. To date, however, very few have addressed future changes in terms of probability of occurrence, and none have quantified the evolution of rare nonstationary hydrological drought characteristics under different warming conditions. This study investigates future changes in the hydrological drought characteristics under varying global warming levels (1.5°C, 2°C, and 3°C), using nonstationary extreme value analysis combined with a Bayesian uncertainty framework across 200 river catchments in the UK. The analysis utilizes the enhanced future Flows and Groundwater (eFLaG) dataset, which is based on the most recent UKCP18 climate projections, and incorporates outputs from four hydrological models (G2G, PDM, GR4J, and GR6J). The findings indicate that rising temperatures will significantly influence future drought duration, severity, and intensity across a majority of catchments, with rare droughts (return period of 100-500 years) projected to be more severe in all seasons, particularly in the southern UK. Further, relatively frequent summer droughts (return periods of 10 years) are expected to become shorter but more severe and intense, particularly at higher warming. We observe notable differences between stationary and nonstationary return periods across seasons, with the change becoming more pronounced at longer return periods, particularly for drought severity. Although the trends remain consistent across models under stationary and nonstationary conditions, the results underscore the role of rarity, nonstationarity, and seasonal controls on the future evolution of hydrological droughts in the region.



## 1. Introduction


The recent decades have been some of the warmest on record in the United Kingdom (UK),
and the average land temperature has already increased by approximately 1.2°C compared
to pre-industrial levels (Climate Change Committee, 2021). Many notable drought events
have been recorded in the UK during the periods of 1975-76, 1988-89, 1990-92, 1995-97,
2004-06, 2010-12, and 2022 (Barker et al., 2024; Murphy et al., 2020; Turner et al., 2021).
Projections indicate that by 2050, several regions could face frequent water shortages, driven
by extended spells of hot and dry weather, which are expected to significantly affect river
flows and soil moisture levels (Bevan, 2019). In addition to the adverse impacts of climate
change, the increasing demand will pose water management challenges in the future, which
is particularly crucial for the south-eastern part of the UK, which is expected to experience
more significant changes in the long-term climate (Bevan, 2022). However, droughts are not
only expected to become more frequent, but also more spatially coherent, especially during
the summer season, which could further complicate drought management strategies(Tanguy
et al., 2023b). The growing awareness of drought as a major and increasing hazard and its
impacts has prompted a significant acceleration of research on changing drought risk in the
UK, and significant changes in water resource management practices. In particular, the
OFWAT 'Duty of Resilience' stipulates that water utilities must plan to ensure security of
supply to very extreme events (OFWAT, 2015) in practice, 1:500-year droughts.
Understanding and preparing against these extreme hydrological events is of most societal
importance for the UK due to their disproportionate impacts on water resources, agriculture,
ecosystems, and public health. For instance, the cost of relying on emergency drought
measures in the UK is projected at £40 billion, whereas proactively building water resilience
would cost £21 billion over the same period (National Infrastructure Commission, 2018).
Furthermore, the annual cost to maintain resilience to severe droughts is estimated at £60–
600 million. For extreme droughts, this rises to £80–800 million per year (Climate Change
Committee ,2019).
Given the relative brevity of most hydrological records, the need to ensure resilience to very
rare extremes has prompted the widespread adoption of stochastic simulation methods to
generate long time series from which we can sample such rare events. However, several lim-
itations and complexities arise from using such methods when understanding extreme event



evolution under anthropogenic climate change (Counsell and Durant, 2023; Environment
Agency, 2025). There is therefore merit in directly analysing climate change projections to
assess the changing return levels of events of a given rarity, including those very extreme
events of the most importance for water resources planning. In this study, return levels have
been defined as the values of a variable (here duration, severity, and intensity) expected to
be exceeded on average once every $T$ years, where $T$ is the return period.  However, the
complicated nature of the drought hazard and its relatively infrequent occurrence,  and the
diverse and uncertain spatiotemporal patterns of hydrological droughts make severity and
rarity assessments complicated (Brunner et al., 2021). Further, understanding future changes
in hydrological drought, in particular, remains limited for the UK, as the majority of studies
have primarily focused on analysing changes in drought magnitude between current and fu-
ture periods, using threshold-based metrics rather than exploring the evolving nonstationary
dynamics of various drought characteristics in the future (Barker et al., 2019; Chan et al.,
2022; Kay et al., 2021). More recently, Parry et al., (2024) utilised a newly developed nation-
ally consistent, multi-model ensemble of hydrological projections enhanced future Flows and
Groundwater (eFLaG) dataset (Hannaford et al., 2022a) to quantify future UK hydrological
droughts. The study conducts the analysis for baseline, and future periods as well as transient
changes in low-flows characteristics, but did not consider droughts in a probabilistic sense
and could not therefore shed light on changing likelihood of very rare/extreme events. Also,
there has been a lack of research focusing on understanding the evolution of hydrological
droughts in the UK under different warming conditions (1.5°C, 2°C, 3°C, and so on), which is
very important from a risk planning point of view(Tanguy et al., 2023a). Global warming level
assessments can be used to support timely adaptation of drought management strategies,
inform policy decisions aligned with global targets, and ensure resilience under plausible fu-
ture warming scenarios.
The analysis in most of the previously mentioned research for the UK is based on the analyses
of extreme events relying on the assumption of stationarity, which assumes that the
probability distribution parameters of a drought characteristic remain constant over time (Wu
et al., 2024). However, it is well-accepted that rising temperatures introduce nonstationarity
into hydrological systems, challenging the conventional approaches to drought analysis (Wu
et al., 2024). This nonstationarity might lead to inaccuracies in estimating the return levels of



extreme events for any design return period under evolving climatic conditions. Coles, (2001)
highlighted that assuming stationarity can lead to an underestimation of extreme event
probabilities. Therefore, incorporating nonstationarity, particularly due to rising
temperatures, is crucial for accurately modelling future drought characteristics (Salas and
Obeysekera, 2014). One of the important aspects of probabilistic modelling of extreme
hydroclimatic events is the uncertainty in estimated parameters (Leng et al., 2024).
Traditional methods, such as L-moments (Parvizi et al., 2022), method of moments (Lück and
Wolf, 2016), and maximum likelihood estimation (Jha et al., 2022), typically rely on point
estimates of parameters, without adequately addressing this issue. However, Bayesian
methods have found their utility for addressing these challenges in parameter estimation
processes (Baykal et al., 2024; Liu et al., 2024). This approach allows for obtaining the
posterior distribution of parameters by integrating over the existing parameter space.
Additionally, the introduction of Markov Chain Monte Carlo (MCMC) methodology facilitates
the approximation of integrals by using a Markov chain with the posterior distribution
(Chandra et al., 2015). This paper uses a nonstationary extreme value analysis (EVA)
framework with Bayesian uncertainty assessment to analyse the evolution of future
hydrological drought characteristics in the UK with specifically including rare droughts (return
period >=100 years). Leveraging the benefits of the eFLaG river flow datasets, which comprise
four hydrological models' (GR4J, GR6J, PDM, and G2G) outputs, this study analyses transient,
century-long projections at a daily resolution over 200 catchments in the UK. It examines the
evolution of future hydrological drought characteristics under three different Global Warming
Levels (GWLs): 1.5°C, 2°C, and 3°C, with a particular focus on extreme droughts. By focusing
on a range of warming scenarios, we aim to capture the full spectrum of possible future
hydrological drought conditions under different climatic conditions. In doing so, this study
provides critical insights for policymakers and water resource managers to better understand
and prepare for future hydrological drought risks and their uncertainties under the influence
of climate change.
## 2. Data and methods
### 2.1. eFLaG data set: hydrological models and future river flow projections
This paper utilizes the eFLaG dataset which are nationally consistent and spatially coherent
hydrological river flow projections for the UK based on UKCP18 - the latest climate projections
from the UK Climate Projections programme (Hannaford et al., 2022a; Lowe et al., 2018;



Murphy et al., 2018). The eFLaG dataset encompasses hydrological model simulations of river
flow ('simobs' and 'simrcm') for over 200 catchments in the UK. In this context, 'simobs' refers
to observation-driven simulations (1989-2018), while 'simrcm' denotes outputs generated
from hydrological modelling using 12km UKCP18 RCM (Regional Climate Models) projections
(up to 2080). The 'simrcm' projections comprise a 12-member ensemble generated through
perturbed-parameter runs of Hadley Centre climate models (GCM, HadGEM3-GC3.05) and
RCM (HadREM3-GA705) (Murphy et al., 2018). It should be noted that all 12 ensemble
members originate from the same model framework and are based on the high emissions
scenario (RCP8.5).
GR4J and GR6J, members of the 'airGR' family, are lumped catchment rainfall-runoff models
known for their simplicity and efficient calibration function (Kuana et al., 2024). The
Probability Distributed Model (PDM) offers configurable options for catchment rainfall-runoff
modelling, allowing for various permutations to be tested across catchments (Moore, 2007).
Grid-to-Grid (G2G) is a distributed hydrological model utilized for simulating natural river
flows across Great Britain at a 1km resolution, providing consistent national-scale flow
estimates (Bell et al., 2018). These models have been successfully applied in diverse
hydrological studies, and several publications detail their versatility and wide-ranging
applicability (Kuana et al., 2024; Ndiaye et al., 2024; Tanguy et al., 2023b). Detailed metadata
and site listings are stored and accessible through the Environmental Informatics Data Centre,
which can be referred for more information(Hannaford et al., 2022b). In this study, we have
utilised all 200 catchments for our analysis. For the nonstationary modelling of drought
characteristics for each catchment, we utilised the recently developed CHESS-SCAPE
temperature datasets, which are bias-corrected 1km resolution gridded data also derived
from UKCP18 projections (Robinson et al., 2022a) as a covariate.
### 2.2. *Nonstationary analysis of future drought characteristics*
The impact of adverse climate change effects has prompted scrutiny of the stationary
assumption regarding hydroclimatic variables, leading to heightened interest in the concept
of nonstationarity within the research community. The concept is also pertinent to planners
using projections of hydrological information and data in their decision-making. In this study,
the drought characteristics were fitted with the generalized extreme value (GEV) distribution
with a cumulative distribution function given by Eq. (1) (Coles, 2001):



$$G(x; \mu, \sigma, \xi) = \begin{cases} \exp\left\{-\left[1 + \left(\frac{(x-\mu)\xi}{\sigma}\right)\right]^{-\left(\frac{1}{\xi}\right)}\right\}, \sigma > 0, \quad 1 + \left(\frac{(x-\mu)\xi}{\sigma}\right) > 0, \xi \neq 0 \\ exp\left\{-\exp\left[-\frac{x-\mu}{\sigma}\right]\right\}, \sigma > 0, \xi = 0 \end{cases} \qquad (1)$$

Here, $\mu, \sigma$ and $\xi$ are the location, scale, and shape parameters of the distribution. Daily

temperature anomaly ($\Delta T$) from the CHESS-SCAPE data (Robinson et al., 2022a) was selected

as the covariate to quantify the temperature-dependent signals for future river flow.

The incorporation of linear dependency in the location parameter is a common practice in

nonstationary modelling, and similar applications to the scale parameter have been

advocated by Yilmaz and Perera, (2014). However, Gilleland and Katz, (2016) argue against

introducing covariates solely to the scale parameter without corresponding variations in the

location parameter. Further, the estimation of the shape parameter under a time-varying

framework is challenging due to the uncertain tail behaviour of the distribution, especially in

limited data settings, and is therefore often kept constant (Ragulina and Reitan, 2017). In our

study, only the location parameter for historical and future streamflow extremes was

assumed to be a linear function of temperature. Hence, the parameter set takes the form of

$\mu(t) = \mu_0 + \mu_1 c(\Delta T), \sigma(t) = \sigma$ and $\xi(t) = \xi$. Parameter estimation was conducted utilizing

the maximum likelihood function, chosen for its capability to incorporate nonstationarity into

the distribution parameter (Strupczewski et al., 2001) as given by Eq. (2):

$$L(\theta) = -nlog\sigma - \left(1 + \frac{1}{\xi}\right)\sum_{i=1}^{n}\log\left[1 + \xi\left(\frac{x_i - \mu}{\sigma}\right)\right] - \sum_{i=1}^{n}\left[1 + \xi\left(\frac{x_i - \mu}{\sigma}\right)\right]^{\left(-\frac{1}{\xi}\right)}, 1 + \xi\left(\frac{x_i - \mu}{\sigma}\right) > 0 \qquad (2)$$

Here, $L(\theta)$ is the likelihood function of the parameter vector $\theta$ and $n$ is the sample size. By

minimizing the above function, the distributions of parameters for both stationary and

nonstationary cases were formulated. The comparative statistical significance of stationary

and nonstationary models was assessed by using the likelihood ratio test (L.R. test) (Posada

and Buckley, 2004) which is derived using Eq. (3):

$$2\left[nllh_s - nllh_{(NS)}\right] > c_\alpha \qquad (3)$$

Here, $nllh_s$ and $nllh_{(NS)}$ are the negative log-likelihood values of stationary and

nonstationary models. Further, $c_\alpha$ represents the $(1 - \alpha)$ quantile of the Chi-square

distribution. The difference between the stationary and nonstationary models is expected to

conform to an approximate chi-squared distribution at a specific significance level α (5% in

this case). The null hypothesis of stationarity is rejected when the p-value exceeds 0.05.






### 2.3. Bayesian framework for parameter uncertainty


As discussed above, parameters for both stationary and nonstationary methods are derived
using the maximum likelihood approach, which only provides point estimates without
accounting for uncertainty. Bayesian analysis aims at updating parameter uncertainty
through a prior distribution using Bayes' theorem (Sarhadi et al., 2016). This approach
combines the prior distribution and the data's likelihood function to form the posterior
distribution, incorporating additional information to enhance predictive modelling. The
posterior distribution is obtained by multiplying the likelihood function by the prior
distribution of the parameter (Eq. 4):
$$p(\Theta \,|x) \; \propto \; p(x|\Theta) \, p(\Theta) \qquad (4)$$

Here, $p(\Theta \,|y)$ denotes the posterior distribution of the parameter vector $\Theta = (\mu, \sigma, \xi)$, $p(\Theta)$
represents the prior distribution, and $p(y|\Theta)$ denotes the likelihood function corresponding
to the GEV distribution evaluated at $x_{i...n}$ where $n$ is the number of observations. We utilised
a non-informative prior distribution for location parameter modelling. Given the complexity
of solving Eq. (4) analytically, numerical methods like MCMC sampling are utilized to produce
numerous realizations from the posterior distribution (Reis and Stedinger, 2005). Further, we
can estimate desired return levels for a given probability of occurrence $(p)$ by employing Eq.

205 (5):

$$Z_p(\hat{\mu}, \hat{\sigma}, \hat{\xi}) = \hat{\mu} - \frac{\hat{\sigma}}{\hat{\xi}} \left\{ 1 - [-\log(1-p)]^{-\hat{\xi}} \right\} \qquad for \; \xi \neq 0 \qquad (5)$$

$$Z_p(\hat{\mu}, \hat{\sigma}) = \hat{\mu} - \hat{\sigma} \log[\log(1-p)] \qquad for \; \xi = 0$$

The Metropolis-Hastings algorithm is used to sample the parameter vector using the specified
prior and likelihood function. It is crucial to monitor the convergence of the MCMC chain to
ensure it accurately represents the posterior distribution. In this study, Heidelberger and
Welch's convergence diagnostic is used to determine the necessary length of each simulation
(Sharma and Mujumdar, 2022).

### 2.4. Analysis of future drought return levels


The whole analysis is set up to calculate the percentage changes in the return level of the
hydrological drought characteristics in the warming level period as compared to the reference
period. The 30-year reference period was 1989-2018, i.e., the available historical period in



the eFLaG dataset. Relative to this reference period, three warming level periods (also 30-
year) were calculated based on the recently developed CHESS-SCAPE temperature data
projections for the UK (Robinson et al., 2022a). In alignment with the objectives and directives
of the Paris Agreement about limiting global warming, a +1.5°C and +2°C rise in temperature
was considered (Jha et al., 2023). Moreover, a warming level of +3°C was also considered,
corresponding to the projected warming expected to be attained by the year 2100 under
existing nationally determined mitigation goals (Seneviratne and Hauser, 2020). The starting
year of each warming level period is defined as the initial year of the 30-year interval wherein
the mean warming exceeds the respective warming level. We considered the last 30-year time
period, in case, the +3°C warming period exceeded the end of the century. For example, in
cases where the warming period is identified as 2080-2110, we instead use the 2070-2100
window to remain within the 21st-century bounds.
To identify hydrological drought events, we used a variable threshold-based approach that
has been widely applied for drought identification (Sarailidis et al., 2019). First, we calculated
the daily mean flows for the reference period eFLaG series. This was done for each of the 12
ensemble members of each of the four hydrological models. The 30-day moving window
centred around each day of the year was calculated for each of the 12 members (for all four
models) and pooled to calculate the daily 90[th] percentile exceedance flow (Q90). Hence, 365
Q90 thresholds (one for each day of the year, assuming 365 days) were derived for the
baseline period. A catchment was considered to be in drought on any given day when the
flow dropped below the baseline Q90 threshold for that day. A pooling procedure across
drought events was also applied, where two distinct events separated by a single day were
combined into a single drought event, provided the magnitude above the threshold did not
exceed the accumulated deficit before this single day. To avoid uncertainty arising due to non-
significant drought events, we excluded those with a standard duration of less than 30 days.
Figure 1 schematically represents the derivation of drought characteristics using the variable
threshold method and a flow chart of the methodology used. Having identified individual
events, three event characteristics were computed for each season (i.e. winter: December-
February, spring: March-May, summer: June-August and autumn: September-November)
which are duration - the number of days over which a drought occurs, severity - the
accumulated flow deficit across all days, and intensity - the ratio of drought severity and
duration of a drought event.






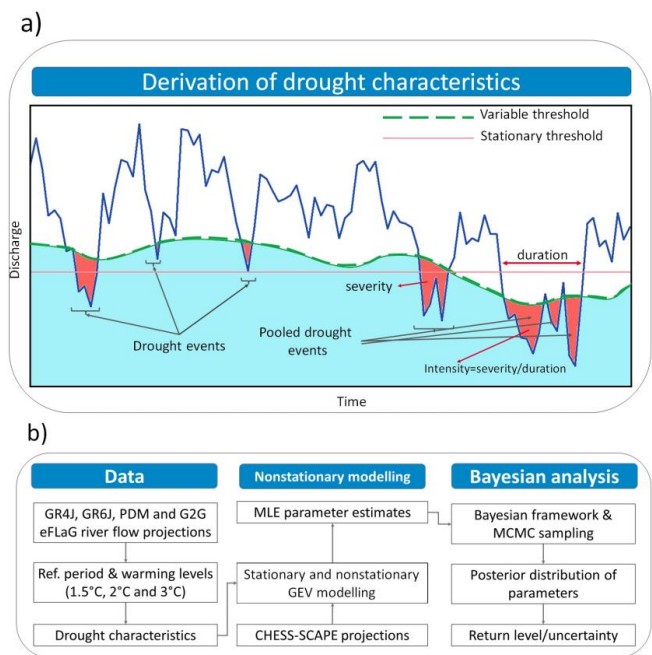

**Figure 1.** a) Variable threshold methodology used to identify and characterise drought events,
b) Methodological framework utilized in the analysis.


## 3. Results and discussion

### 3.1. Nonstationary properties and Bayesian parameter estimates

Once the drought characteristics for all four models across all four seasons were calculated,
the nonstationarity was assessed using the likelihood ratio test. Figure 2 represents the
percentage of nonstationary catchments for each drought characteristic across three
warming levels and seasons. It shows that the nonstationary properties of catchments
depend on the combination of the drought event characteristics, warming levels, and
seasons. Future hydrological drought duration is found to be nonstationary in most
catchments across warming levels and seasons. This is most noticeable at 3°C warming, where
almost all catchments across seasons are depicting nonstationarity in future hydrological
drought duration. Interestingly, future drought intensity at lower warming levels appears to
be stationary. Only during the winter season does drought intensity exhibit a trend of rising
nonstationarity as the warming increases. Further, at least half of the catchments display
nonstationary hydrological drought severity characteristics across warming levels, except




during the summer season at lower warming levels. The trend across models remains overall
similar, and no noticeable difference in the ability to capture nonstationarity was observed.
However, the changes in nonstationary properties, their dependence on warming conditions,
characteristics, and seasons need consideration while modelling the evolution of future
hydrological droughts.

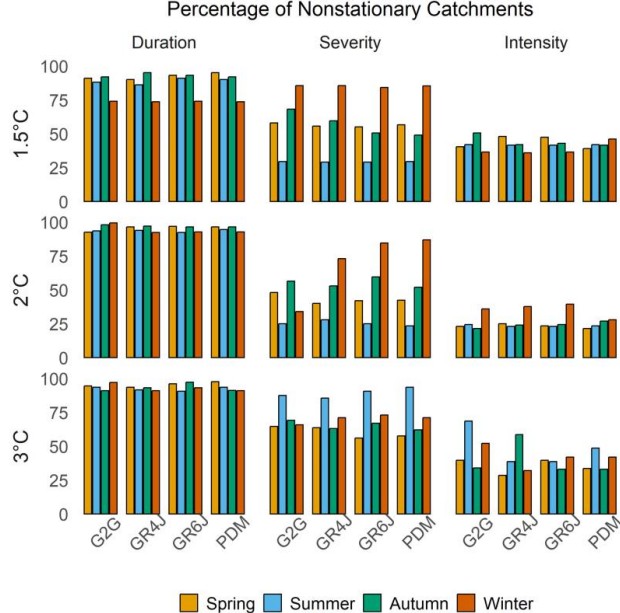

**Figure 2.** Percentage of nonstationary catchments for each event characteristics, hydrological
models and warming levels.


Once the nonstationarity was assessed, we derived the parameter distribution for calculating
the return levels of future and historical drought duration, intensity, and severity. Figure 3
demonstrates the mean and standard deviation of the posterior distribution of parameters
obtained using the Bayesian framework for the GR4J model during the summer season at
+3°C. The spatial distribution of parameter means and standard deviation, particularly for
duration, suggests that there is relatively higher uncertainty in the location parameter in the
south-eastern catchments. The south-east not only experiences a higher magnitude of mean
location parameter but also higher uncertainty which is in agreement with previous studies
depicting more significant changes in future drought conditions in this region (Kay et al.,
2021). The variation of the location parameter across catchments for drought intensity and
severity exhibits more or less similar behaviour. It can also be observed that catchments with



a higher magnitude of the location parameter exhibit a higher standard deviation. This is
crucial and calls for more caution as it denotes, for e.g., a catchment with a higher duration
of drought might show higher uncertainty in the estimates. We also demonstrate the
robustness of the employed method by comparing the curves of posterior distributions of
location parameters for a sample catchment (Dee in Scotland, NRFA ID: 67018) for the
reference period and +3°C warming (Figure S1). The location parameter for future drought
duration shows a lower value, whereas intensity and severity are generally higher. This
pattern is consistent with the findings from the return level analysis, which are presented in
the next sections. Figure S1 also shows that the possible spread of location parameters for
future drought characteristics is well constrained. This is critical as it ensures that the model
provides robust estimates of parameters, especially for understanding future changes in
drought characteristics under projected warming.

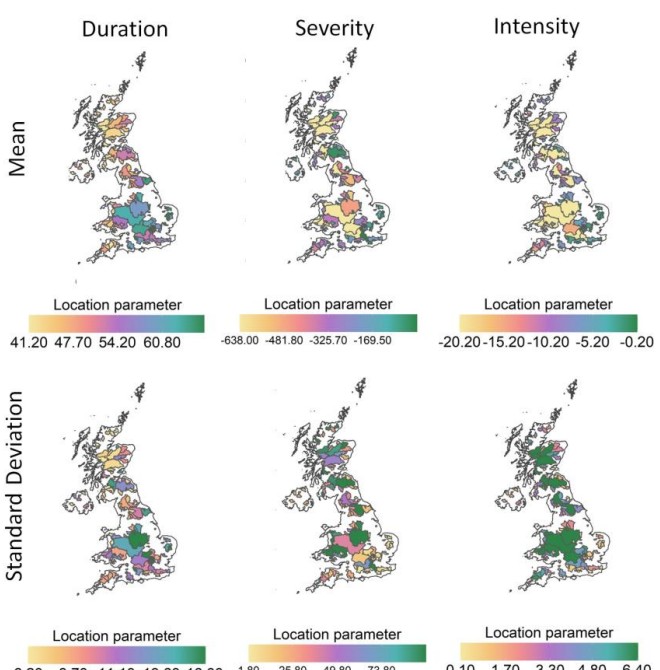

**Figure 3.** Mean and standard deviation of parameter samples for GR4J model during summer season at 3°C warming level.




### 3.2. Return levels of different drought characteristics

Next, we calculated the return levels of drought duration, severity and intensity at different return periods (10, 100, and 500 years) using parameter samples from the posterior distribution obtained through Bayesian analysis. The return levels were calculated for both the reference period and the warming level periods, considering the stationary case as well as nonstationary case. The results presented in the main text of this paper focus exclusively on the mean return levels; however, different return levels corresponding to median, 75th, and 25th quantiles of the posterior parameter distribution were also calculated and can be referred to in the supplementary information (Figure S2a-c) for more insights about uncertainty in the estimates.

Figure 4a, b shows the model average percentage change in mean nonstationary return levels for 10-year (frequent droughts) and 500-year (rare droughts) return levels, respectively. The return level is dependent on the rarity of the drought, as changes in return levels are more pronounced for a 500-year drought compared to a 10-year drought, with the former exhibiting more distinct spatial characterisation. The spatial distribution of percentage changes in the mean 100-year return level is shown in the supplementary information (Figure S2, S3, S4). For drought duration, the overall return levels are expected to be higher for 500-year droughts during the autumn and winter seasons, whereas they are expected to be lower for 10-year droughts in the same seasons. This increase in the risk of prolonged extreme droughts in autumn and winter is concerning, given that the winter half-year is the critical time for replenishment of aquifers (in the south-east) and reservoirs(Barker et al., 2019; Environment Agency, 2011). The shorter duration of 10-year droughts may slightly ease water stress during more frequent droughts in these seasons however, any potential benefits could be offset by increased drought intensity, making the overall water management plan in the country still challenging. In Fig. 4b, which shows longer drought durations, regions in the north and west, which rely almost entirely on surface water and lack the buffering capacity of groundwater, might be significantly affected, whereas areas in the south-east dominated by groundwater-fed systems might experience delayed drought impacts, offering a degree of resilience during prolonged dry periods. Previous studies have also shown significant variability in hydrometeorological drought characteristics, both in the current period and in future projections, specifically in the southern part of the country (Barker et al., 2019; Di Nunno and Granata, 2024; Reyniers et al., 2022). Compared to intensity, duration return




levels have more distinct regional attributes for rare droughts - particularly in the spring and
summer season where some of the catchments show abrupt negative changes in return
levels. Studies suggest that the UK is likely to experience warmer and wetter winters alongside
hotter and drier summers in the future(Lowe et al., 2018).

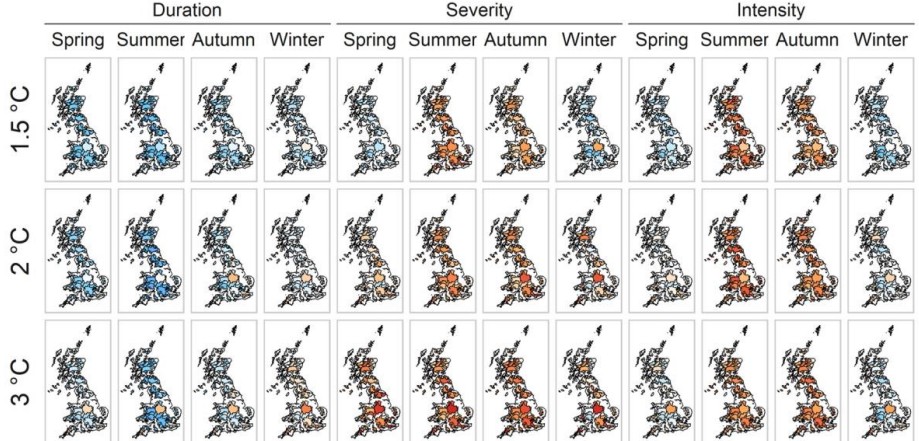

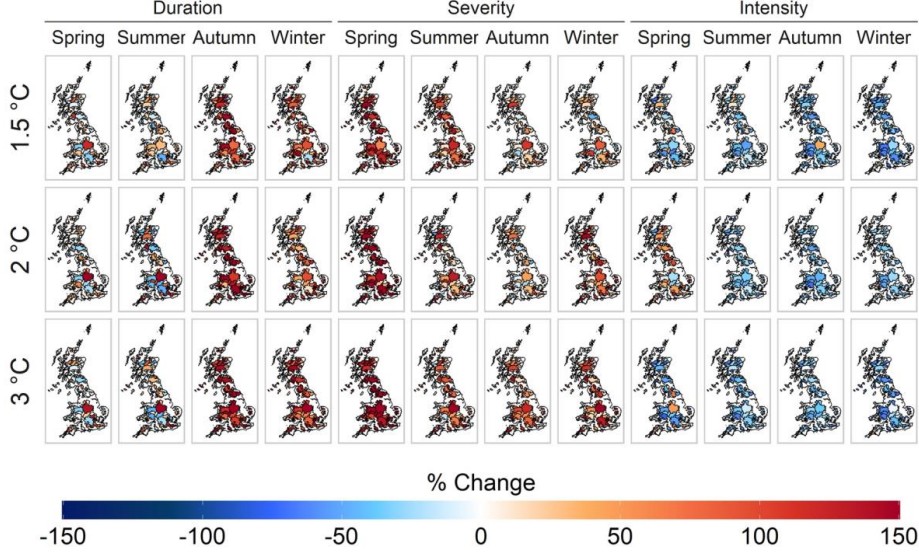

**Figure 4 a, b.** Percentage change in mean nonstationary a) 10-year and b) 500-year return levels for different drought characteristics across all warming levels and seasons.




Additionally, most projections indicate an overall increase in potential evapotranspiration,
with seasonal variations in the rate of change, but a consistent upward trend on an annual
basis (Robinson et al., 2022b). This could be one of the possible drivers of longer future
drought durations for frequent droughts or higher severity of rarer droughts, particularly in
the summer season (Kay et al., 2020; Murphy et al., 2018). Future severity is observed to be
increasing for both frequent and rare droughts in most catchments, except during the winter
season for frequent droughts at lower warming levels. Season-wise, the increasing changes
in the severity of rare droughts in the spring are highest, followed by summer, winter, and
autumn. This increase is more substantial at higher warming levels, which indicates that both
rare and frequent droughts are, in general, expected to be more severe in the future under
the influence of rising temperature (Parry et al., 2024). Further, the intensity of droughts with
a 10-year recurrence interval is projected to increase during the autumn and summer
seasons. Conversely, the intensity of droughts with a 500-year return period is found to be
decreasing in most seasons across all warming levels. It should be noted that we have
considered the mean intensity, which is a function of both duration and severity, and highly
intense frequent droughts in the future, particularly in autumn and summer seasons, could
be due to highly severe droughts over a smaller duration (Figure 4a).

**3.3.    Difference between stationary and nonstationary return levels**
To understand the role of temperature in governing changes in future drought characteristics,
we compared the stationary return levels with the nonstationary return levels. Figure 5a,b
shows the distribution of model-average percentage change in nonstationary and the
stationary return levels for seasons and warming levels. The difference in percentage change
in hydrological drought intensity return levels for the stationary and nonstationary cases is
negative, particularly for higher return periods and warming levels across seasons. This might
be because most catchments for drought intensity exhibit stationary characteristics (Figure 2)
and show similar spatial patterns for stationary return levels as well (Figure S3a-c). For
drought severity, the changes in return levels tend to show a decreasing trend with increased
rarity. However, this is exclusive to the autumn season as drought severity in other seasons
exhibits higher return levels with higher return periods of droughts. Similar results were
observed for the stationary return levels; however, while the overall trend remains
consistent, there is a significant difference in the magnitude of the stationary and



nonstationary return levels. Figure S3a-c in the supplementary information shows the spatial
patterns of stationary return levels.

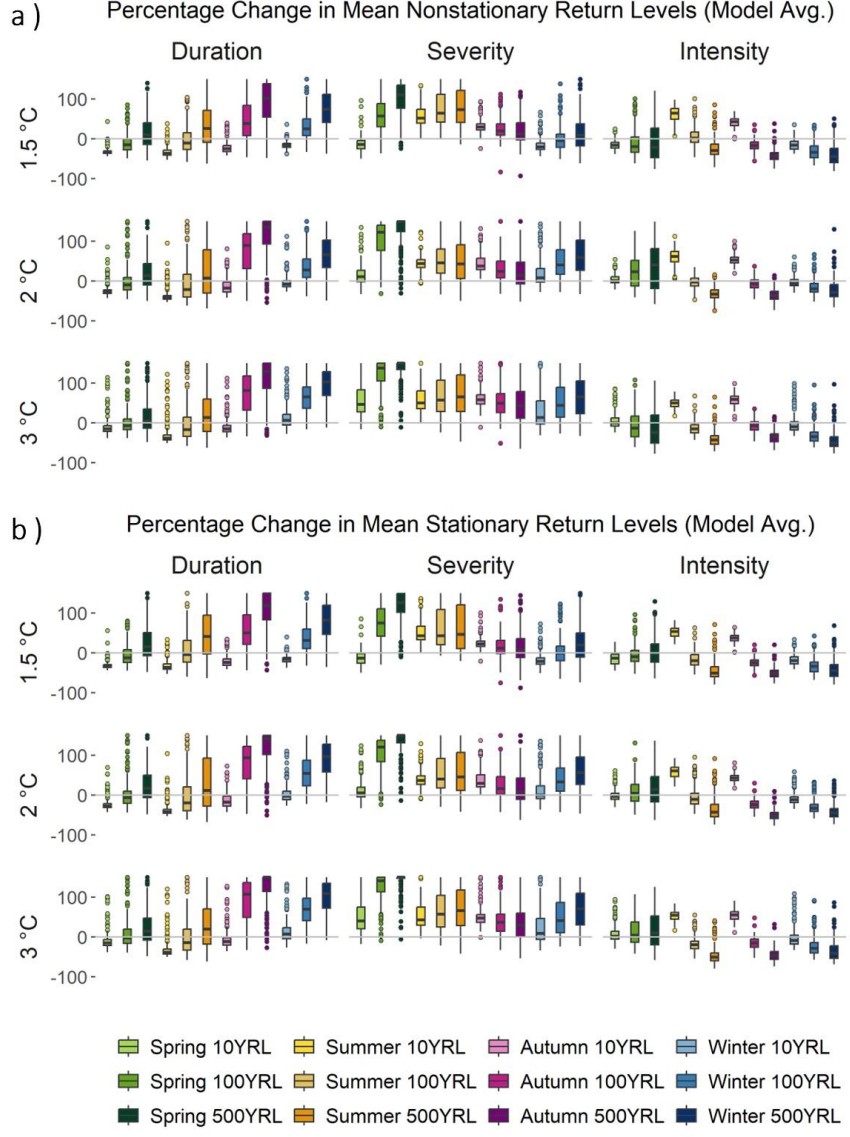

**Figure 5 a, b.** Percentage change in mean c) nonstationary and d) stationary return levels (10,100 and 500 years) for different drought characteristics across all warming levels and seasons.


The incorporation of 100-year return levels also confirms the trends in the results, showing
that as droughts become less frequent, the changes in return levels become more





pronounced. It can also be concluded that rarer droughts are not only accompanied by larger-
scale changes in return levels but also by larger variability. This heightened variability
underscores the need for robust modelling approaches to better understand the impacts of
rare hydrological droughts in the UK under climate change. Most previous studies in the UK
have considered different climate model outputs or hydrological models but did not take into
account the variability induced due to warming on different drought events on the seasonal
scale (Parry et al., 2024; Rudd et al., 2019). Therefore, the results of this analysis provide more
comprehensive insights into the varying uncertainty of future return levels.
**3.4.  Inter-model differences in return levels**
Further, Figure 6 shows the magnitude of the difference between the percentage changes in
nonstationary and stationary return levels for 3°C warming level.  Results are shown for each
model to demonstrate the variability among models. The difference between the
nonstationary and stationary return levels is smaller for drought intensity compared to
drought duration and severity. This outcome was expected due to the relatively lower level
of nonstationarity detected in the drought intensity projections (Figure 2) and a higher
severity and lower duration compared to the reference period (Figure 4a,b). This suggests
that the mean flow deficit relative to the historical drought threshold on any given day in the
future is less likely to be related to temperature change than for duration and severity.
However, the number of days over which drought might occur and the total accumulated flow
deficit across all days of a drought are more likely to be affected by these factors at higher
warming levels. Moreover, the duration of more frequent droughts being less affected by
rising temperatures is also confirmed by minimal difference between stationary and
nonstationary return levels across seasons, which changes significantly when higher return
levels are considered (Figure 6).
Overall, the results indicate that failing to incorporate temperature effects in modelling
duration for longer return period droughts can lead to significant uncertainty regarding their
future return levels. This underestimation and variability are most amplified for future
drought severity, where it is evident that temperature influences across models, seasons, and
warming levels might lead to more severe droughts.  To further confirm this, we analysed the
distribution of the 25th, 75th quantiles, and the median return levels for different warming
levels (Figure S4a-f), which shows a similar trend. Further, assessing model performance for
future periods compared to a baseline period is challenging because different hydrological



models capture processes and uncertainties based on their individual structure and
operational specifications. Therefore, it is important to incorporate multiple models for more
confident estimates of future changes in drought characteristics (Hannaford et al., 2023; Lane
et al., 2022). In this setting, with four hydrological model outputs assessed, for each drought
characteristic, the return levels across the UK are primarily driven by the rarity of the event
in different seasons rather than the model itself.

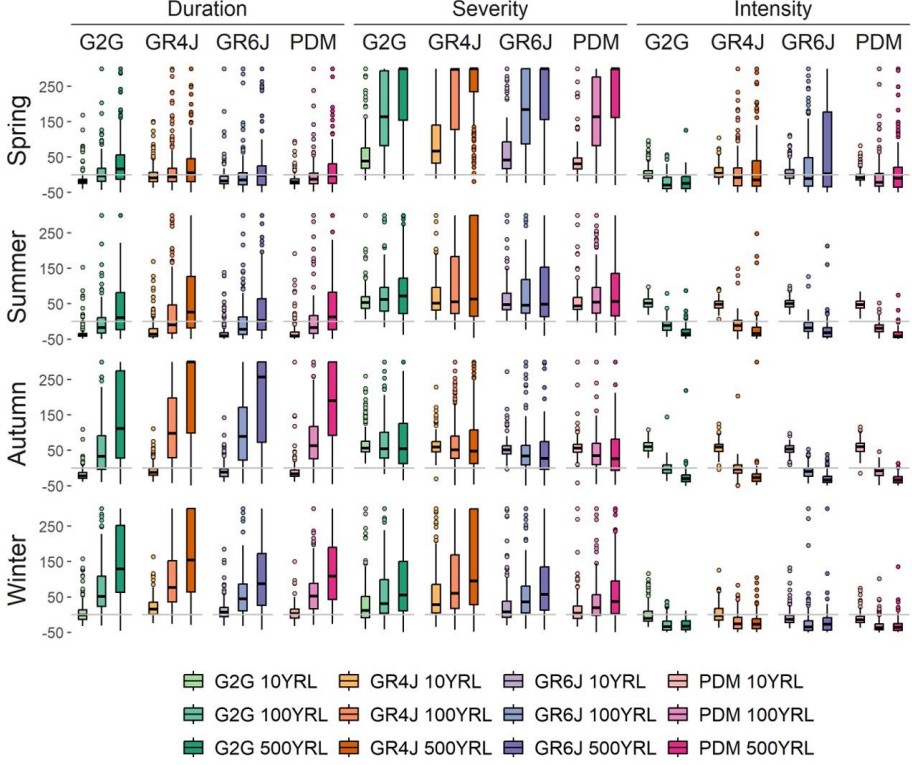

**Figure 6.** Difference in percentage change in return levels for mean nonstationary and stationary return levels for different drought characteristics across all seasons and 3°C warming levels.


Although the results from this analysis are consistent across the hydrological models, a more
detailed uncertainty partition analysis could be conducted in the future to gain a deeper
understanding of the inter-model differences in the projected characteristics of future
droughts. Further studies could also incorporate catchment hydrometeorological
characteristics in the nonstationary modelling set-up to understand the role of changing



catchment conditions in governing the drought characteristics. In this study, we have looked
at the drought characteristics independently, however, the dependence of drought
characteristics over time, as well as their evolution in a compound setting could give more
useful insights about their interrelation in the future.  Despite this, the findings from this
analysis give crucial insights about the changing future hydrological drought characteristics in
the UK under climate change.  The results not only point out changing magnitudes of drought
duration, severity, and intensity but also provide robust estimates of uncertainty on different
spatial and temporal scales, which can be considered while designing more targeted and
localized strategies against drought-related challenges in the future.
**4. Conclusions**
This study attempts to understand the evolution of future hydrological droughts in the UK
under different warming conditions, utilising nonstationary extreme value analysis with a
Bayesian framework for parameter uncertainty. We used the recently developed eFLaG
projections to investigate changes in drought characteristics in terms of return levels. The
findings indicate that future temperature changes contribute significantly and uniquely to
hydrological droughts' characteristics - duration, severity, and intensity.  Results demonstrate
that the future changes in these characteristics are highly dependent on the season and the
rarity of droughts. Drought severity in most cases, irrespective of rarity and season, appears
to be increasing in the future at higher warming levels. However, future drought duration and
intensity are showing both increasing and decreasing trends depending on the season and
return period of droughts. This also underscores the varying degrees of nonstationarity
exhibited by different drought characteristics, which should be carefully considered while
planning measures against future drought risks in the UK. The projected return levels,
particularly for rare and high-impact events, also show a higher level of uncertainty in their
magnitude as compared to more frequent events, which can be critical for risk management
and adaptation strategies. Overall, this research underlines the importance of considering the
influence of temperature-induced nonstationarity in modelling future changes in hydrological
drought characteristics. Results from both stationary and nonstationary cases across different
seasons, rarities, and warming levels provide comprehensive insights that can be utilised by
policymakers and water managers to develop effective strategies against future risks.




**Code and data availability**

The eFLaG river flow projections analysed in this study are stored at the UKCEH's Environmental Information Data Centre and can be freely accessed as DOI datasets. Please ensure these data are cited in full when used in any application: https://catalogue.ceh.ac.uk/documents/1bb90673-ad37-4679-90b9-0126109639a9. The CHESS-SCAPE dataset can be downloaded from the NERC Environmental Data Service (EDS) Centre for Environmental Data Analysis (CEDA) via the following link: https://doi.org/10.5285/8194b416cbee482b89e0dfbe17c5786c. The R scripts used for analysis were developed using publicly available packages, such as 'extRemes', 'evir', 'coda', 'foreach', and 'doparallel', which support extreme value analysis, Markov Chain Monte Carlo diagnostics in a parallel environment.

**Author contribution**

Conceptualization was done by SJ, JH, MT, and LB. Methodology development and analysis were carried out by SJ. The original draft was written by SJ and JH. Reviewing and editing of the manuscript were performed by LB, JH, and MT. Supervision of the work was provided by JH, LB, and MT.

**Competing interests statement**

The authors declare that they have no conflicts of interests.

**Acknowledgements**

We also acknowledge the use of the JASMIN high-performance computing facility for the Bayesian analysis conducted in this study. JASMIN facility is operated by the Science and Technology Facilities Council on behalf of the Natural Environment Research Council.

**Financial support**

This work has been financially supported by the Hydro-JULES Programme (NE/S017380/1) funded by Natural Environment Research Council.



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
