# Peer review of "Evolution of nonstationary hydrological drought characteristics"

_EGUsphere, 2025_

## Author Comment (AC1)

**Reply to Comments by Referee #1**

**Title:** Evolution of nonstationary hydrological drought characteristics in the UK under warming

**Recommendation:** Accept after corrections

**Reply:** The authors would like to thank the reviewer for carefully reviewing the manuscript and providing valuable comments and suggestions. Below, we provide point-wise responses to each comment along with the proposed changes in the revised version of the manuscript.

**Comment 1:** Explain the non-stationarity in the hydrological drought time series. How are the future groundwater estimates calculated and how accurate it is?

**Reply 1:** Non-stationarity in the river flow time series has been assumed in a statistical sense. Specifically, the location parameter of time series is allowed to vary as a linear function of the corresponding temperature anomaly time series. This approach captures long-term shifts in the mean hydrological conditions driven by warming which is one of the main aims of the work.

In this study we have considered the enhanced future Flows and Groundwater (eFLaG) dataset which are nationally consistent hydrological projections derived from a range of hydrological models (Grid-to-Grid, PDM, GR4J and GR6J) and groundwater recharge model ZOODRM (zooming object-oriented distributed-recharge model) (Hannaford et al., 2022). However, in this paper we have only focussed on the river flow projections for our analysis and did not consider the groundwater data. We propose to add a more detailed clarification about this in the manuscript in Section 2.1.

**Comment 2: Line 35:** 1.2 deg. is the how many years average?

**Reply 2:** For more clarity, and to include a UK-specific and up-to-date reference, we will replace the statement in **Line 34-36** with the following:

With ongoing climate change and global warming, the United Kingdom (UK) is experiencing a pronounced warming trend, with the most recent decade (2015-2024) averaging 1.24 °C above the 1961-1990 baseline (Kendon et al., 2024).

**Comment 3: Line 80-81:** …."transient changes in low-flows characteristics"; what is the meaning of the transient changes here?

**Reply 3:** Thank you for the comment. We propose to modify **Line 80-81** for more a clearer explanation as following:

Hannaford et al. (2022a) utilized the daily eFLaG dataset, which consists of transient time series (continuous daily data from 1980 to 2080), to explore changes in drought characteristics. These transient analyses capture how river flows evolve over time, rather than only comparing baseline and future time slices. However, they do not account for the probabilistic assessment of droughts or changes in their likelihood under future warming.

**Comment 4: Line 113:** What is "transient" is not clear from the introduction?

**Reply 4:** Here, "transient" refers to a continuous time series that evolves over time (in this case daily data), rather than discrete baseline and future periods. We will add this information in the introduction as well as make it clearer in Section 2.1. which discusses eFLaG data in detail.

**Comment 5: Line 117-118:** "we aim to capture the full spectrum of possible future hydrological drought conditions under different climatic conditions."

**Reply 5:** We assume that more explanation is needed on this point. In the paper, the phrase "we aim to capture the full spectrum of possible future hydrological drought conditions under different climatic conditions" refers to our approach of analysing drought characteristics across multiple Warming Levels- 1.5°C, 2°C, and 3°C. By considering a range of warming scenarios, the study examines how the duration, severity, and intensity of droughts may evolve under different climate conditions which are defined by these warming levels.

**Comment 6: Line 133 – 135:** "It should be noted that all 12 ensemble members originate from the same model framework and are based on the high emissions scenario (RCP8.5)." How it is same model framework? Please rephrase or write proper explanation for these lines?

**Reply 6:** We agree that this line requires a more detailed explanation. Thank you for your suggestion. We will modify **Lines 131–135** as follows to provide a clearer and more detailed description:

The 'simrcm' projections consist of a 12-member ensemble generated using perturbed-parameter runs of the Hadley Centre global climate model (GCM, HadGEM3-GC3.05) and regional climate model (RCM, HadREM3-GA705) (Murphy et al., 2018). Each ensemble member represents a plausible variation in model parameters to capture uncertainty in the climate response, while all members share the same underlying model framework and follow the high-emissions scenario (RCP8.5). The 12-member RCM perturbed-parameter ensemble is therefore valuable for representing parameter uncertainty; however, because all members are based on the same model structure and emissions scenario, they do not capture the full range of climate or scenario uncertainties.

**Comment 7: Line 148:** "recently developed CHESS-SCAPE" what do you mean by "recently developed"

**Reply 7:** Thank you for raising this point. We added this given the data publication in 2022 (Robinson et al., 2022), however we agree that the term "recent" is subjective and can vary over time. We will remove this word from **Line 148.**

**Comment 8: Figure 1:** which approach is more suitable for the drought identification; variable threshold or stationary approach? Have authors included the justification and applicability for these methods?

**Reply 8:** In this paper, we have employed a variable threshold method to calculate drought events and did not use constant threshold. The methodology is explained in detail in Lines 230-244. However, we propose that following points in this section will be included for a clearer justification about the applicability of the chosen method after **Line 244:**

The variable threshold method is considered a more suitable and increasingly popular approach compared to the constant (stationary) threshold method for defining hydrological droughts (Anderson et al., 2025; Brunner & Chartier-Rescan, 2024). This method allows for smooth intra-annual variability and identifies drought events when flows fall below the historically expected level on a given day, which would be overlooked by a constant threshold. By considering individual flow percentiles for each day of the year, this approach ensures that a deficit is identified only when flows are significantly lower than the historical flow for that specific time of year, thereby avoiding the misclassification of droughts. Furthermore, the use of pooling procedures and running-window methods ensures that shorter drought events are also correctly captured.

**Comment 9: Conclusions:** Authors are advised to write the conclusions with the focus on the comparative analysis of the drought occurrence in the baseline and future periods under different warmings. Which can help in the framing/modifying the policy for the future dryness events.

**Reply 9:** Thank you for the comment, we agree that more points related to results obtained for baseline period and future warming under different warmings would be helpful for policymaking purposes. We will modify the Section 4. (Conclusions) accordingly and include the following points:

Based on this comparative analysis, we conclude that the most critical policy considerations for future hydrological droughts will revolve around adapting to projected nonstationary changes in the nature of risk. For instance, the finding that drought severity consistently increases across the majority of catchments under higher warming dictates that policy reviews of water resource infrastructure and management plans are necessary to create buffers against larger future deficits. Furthermore, the observation that changes in drought duration and intensity are highly dependent on the season points toward a required shift from uniform, year-round planning to seasonally specific risk management strategies. Finally, the higher uncertainty observed, particularly for rare high-impact droughts, indicates that future policy must explicitly integrate the possibility of extreme outcomes beyond currently accepted limits of uncertainty, requiring robust, nonstationary modelling in all risk management and adaptation strategies.

**REFERENCES:**

Anderson, B. J., Muñoz-Castro, E., Tallaksen, L. M., Matano, A., Götte, J., Armitage, R., ... & Brunner, M. I. (2025). What is a drought-to-flood transition? Pitfalls and recommendations for defining consecutive hydrological extreme events. *Hydrology and Earth System Sciences*, *29*(21), 6069-6092.

Brunner, M. I., & Chartier-Rescan, C. (2024). Drought spatial extent and dependence increase during drought propagation from the atmosphere to the hydrosphere. *Geophysical Research Letters*, *51*(6), e2023GL107918.

Hannaford, J., Mackay, J., Ascot, M., Bell, V., Chitson, T., Cole, S., Counsell, C., Durant, M., Facer-Childs, K., and Jackson, C.: Hydrological projections for the UK, based on UK Climate Projections 2018 (UKCP18) data, from the Enhanced Future Flows and Groundwater (eFLaG) project, Environmental Information Data Centre. doi, 10, https://doi.org/10.5285/1bb90673-ad37-4679-90b9-0126109639a9, 2022

Kendon, M., Doherty, A., Hollis, D., Carlisle, E., Packman, S., McCarthy, M., ... & Sparks, T. (2024). State of the UK Climate 2023. *International Journal of Climatology*, *44*, 1-117**.**

Murphy, J. M., Harris, G. R., Sexton, D. M. H., Kendon, E. J., Bett, P. E., Clark, R. T., Eagle, K. E., Fosser, G., Fung, F., and Lowe, J. A.: UKCP18 land projections: science report, 2018.

Robinson, E. L., Huntingford, C., Shamsudheen, S., and Bullock, J.: CHESS-SCAPE: Future projections of meteorological variables at 1 km resolution for the United Kingdom 1980-2080 derived from UK Climate Projections 2018, NERC EDS Centre for Environmental Data Analysis, https://doi.org/http://dx.doi.org/10.5285/8194b416cbee482b89e0dfbe17c5786c, 2022

---

## Author Comment (AC2)

**Reply to Comments by Referee #2**

**Comment 1:** The paper is well-written and clearly mentions the research gap. It would be more interesting if the authors kept the objective as a point order (**consistent**) before concluding the introduction, so that the reader remains clear about the objective before moving on to the methodology.

**Reply 1:** We thank the reviewer for the careful reading and for the constructive suggestion regarding the structure of the introduction. We agree that listing the objectives in a clear point-by-point format before concluding the introduction will improve reader's clarity about the work before they move ahead. We will include the point-wise research objectives in the last part of introduction and make sure that the following points are incorporated:

In summary, the objectives of this study are as follows:

1. To investigate the projected changes in key hydrological drought characteristics (duration, severity, and intensity) across 200 UK catchments under three future warming scenarios.
2. To apply and compare results from nonstationary and stationary EVA using a Bayesian framework to quantify the role of nonstationarity in governing future hydrological drought risks.
3. Understanding the future evolution of hydrological drought characteristics in UK, specifically for rare events with robust estimation of uncertainty.

**Comment 2:** The author highlighted the role of seasonal controls in future drought. In lines 45 and the abstract, it has been highlighted, but it is unclear whether it is also considered an objective of the study. However, it's clear from the results section onwards, and this is indeed very interesting. However, the authors have not highlighted the importance of seasonality in relation to drought in the research gap and have **not cited any relevant literature**.

**Reply 2:** Thank you for the raising the important point and mentioning that seasonal results are interesting and worth highlighting. We have tried to discuss the season-based results subsequently in detail as we have set the whole analysis on seasonal scale. As mentioned in response to your previous comment, we agree that this point will be highlighted in the revised manuscript as one of the objectives of the study and relevant literature will also be cited at **Line 45-46** as follows:

…drought management strategies (Tanguy et al., 2023b). River-flow projections in the UK are known to be sensitive to seasonal variations in precipitation and potential evapotranspiration, owing to their influence on the seasonal wetting and drying cycles of the soil (Parry et al., 2024). Chan et al. (2024) further highlighted that the likelihood of experiencing a summer month drier than the historically driest recorded month is expected to rise with future warming in certain regions of UK. Although these studies highlight the importance of seasonal controls on UK droughts, a comprehensive probabilistic analysis of drought return levels across characteristics and warming levels is still needed.

**Comment 3:** Can the author **justify why they have selected the 90th percentile threshold**? A suggestion to check with an 80th percentile threshold.

**Reply 3:** We selected the 90th percentile (Q90) threshold to ensure that the analysis captures instances characterised by extremely low historical flows. This choice allows us to focus on severe low-flow anomalies that are hydrologically meaningful, rather than relatively normal variations in streamflow. The Q90 threshold has also been widely used in previous hydrological drought assessments, providing

both consistency and comparability with earlier studies (e.g., Hasan et al., 2020; Prudhomme et al., 2014; Janicka-Kubiak 2025). Furthermore, Q90 is sufficiently stringent to minimise the influence of short-term fluctuations, ensuring that the identified drought events represent genuine low-flow conditions rather than transient anomalies. An additional motivation for adopting the Q90 threshold is our emphasis on addressing uncertainties associated with estimating rare drought characteristics. Using a high-percentile threshold such as Q90 demonstrates that the methodology is robust for detecting extremely low-occurrence drought events, thereby supporting the reliability of our drought characterisation approach.

In the revised version of the manuscript, we will incorporate these clarifications in Section 2.4 (**Line 235**) to make our rationale for selecting the 90th percentile threshold clearer and more transparent. Further, as per your suggestion we will also check an additional percentile threshold as suggested to demonstrate the consistency in analysis and include the results.

**Comment 4:** The majority of studies have considered the drought identification (threshold) on the total observation period, so using a baseline period here is really questionable. Please **justify**. For future data, the threshold should also vary, so the entire period should be considered instead of just the baseline.

**Reply 4:** Thank you for the comment. A major aim of this study is to assess nonstationary changes in drought characteristics relative to a baseline period under climate change. Using a baseline threshold and calculating future statistics against it allows us to capture changes due to shifts in the mean, whereas calculating thresholds over the entire period would dampen these changes and risk underestimating the future state of drought. In this study, we identify drought events using a variable-threshold approach, where flow deficits are measured relative to the reference-period Q90 values which results in drought characteristics naturally varying in the future warming level periods. Thus, although many studies compute thresholds over the full period, here we think using a baseline is justified when the goal is to detect absolute changes in drought behaviour relative to historical conditions. The suitability of this approach has been demonstrated in several studies for UK and globally, for e.g., in Parry et al., 2024; Satoh et al., 2022 and Araujo et a., 2025.

**Comment 5: Line 146** R**epetition** about the number of catchments may be deleted, as it has been clearly stated beforehand.

**Reply 5:** We thank the reviewer for noticing this. The repeated mention of the number of catchments will be removed in the revised version of manuscript.

**Comment 6: Line 186:** Please check if the **null hypothesis** is rejected for a P-value exceeding 0.05; is this the case? Please clarify the exact null hypothesis in accordance with the study and cite the relevant literature. **Present a table which is significant, which is not in the supplementary.**

**Reply 6:** We propose to correct Line 186 and add the following lines including a table in the revised manuscript for more clarification:

The null hypothesis in our study assumes that drought characteristics extremes are stationary, meaning their statistical properties do not change over time or with temperature. Using the likelihood ratio test, this hypothesis is evaluated by comparing the fit of stationary and nonstationary GEV models. The null hypothesis is rejected when the p-value falls below 0.05, indicating that including temperature as a

covariate significantly improves the model. Such an approach is consistent with standard methods in extreme value analysis for hydrological data (Salas and Obeysekera, 2014; Das and Umamahesh, 2017).

**Comment 7: Line 198,** there is an **error in the terminology** used in the expression and in the description.

**Reply 7:** We thank the reviewer for pointing this out. The terminology in **Line 198** will be corrected in the revised manuscript as follows:

$$p(\Theta \,|y) \,\propto\, p(y|\Theta)\, p(\Theta) \tag{4}$$

Here, $p(\Theta \,|y)$ denotes the posterior distribution of the parameter vector $\Theta = (\mu, \sigma, \xi)$, $p(\Theta)$ represents the prior distribution, and $p(y|\Theta)$ denotes the likelihood function corresponding to the GEV distribution evaluated at $y_{i...n}$ where $n$ is the number of observations.

**Comment 8: Line 232:** What is the moving window size with respect to ± how many days?

**Reply 8:** As mentioned in **Line 233**, a 30-day moving window centred on each day of the year was applied.

**Comment 9: Line 240: Cite literature**

**Reply 9: Line 238-241** discusses the pooling procedure applied in the analysis to ensure that closely spaced drought events are treated as a single continuous event, reflecting their cumulative hydrological impact. In the revised manuscript, we propose to cite Parry et al., 2024 and Van loon and Van Lanen 2012 which utilise similar method for the analysis.

**Comment 10: Line 382-385:** Also, the intensity is a function of both duration and severity, so that might be the reason behind the smaller variability?

**Reply 10:** Yes, this difference is smaller for drought intensity as intensity is a derived metric from duration and severity. This results in possible partial compensation of variability due to simultaneous increase/decrease in duration and severity as compared to the case when they are looked at individually.

**Comment 11: Figure 2: Mention unit** for duration, intensity and severity

**Reply 11:** Thank you for the suggestion. We will mention the units for duration, intensity and severity in the revised manuscript.

**REFERENCES:**

Araujo, D. S., Enquist, B. J., Frazier, A. E., Merow, C., Roehrdanz, P. R., Moulatlet, G. M., ... & Nikolopoulos, E. I. (2025). Global future drought layers based on downscaled CMIP6 models and multiple socioeconomic pathways. *Scientific Data*, *12*(1), 295.

Das, J., & Umamahesh, N. V. (2017). Uncertainty and nonstationarity in streamflow extremes under climate change scenarios over a river basin. *Journal of Hydrologic Engineering*, *22*(10), 04017042.

Hasan, H. H., Mohd Razali, S. F., Muhammad, N. S., Mohamed, Z. S., & Mohamad Hamzah, F. (2020). Assessment of probability distributions and minimum storage draft-rate analysis in the equatorial region. *Natural Hazards and Earth System Sciences Discussions*, *2020*, 1-29.

Janicka-Kubiak, E. (2025). Hydrological drought trends and seasonality in selected Polish catchments between 1993 and 2022 using a threshold based approach. *Scientific Reports*, *15*(1), 40454.

Parry, S., Mackay, J. D., Chitson, T., Hannaford, J., Magee, E., Tanguy, M., ... & Wallbank, J. (2024). Divergent future drought projections in UK river flows and groundwater levels. *Hydrology and Earth System Sciences*, *28*(3), 417-440.

Prudhomme, C., Giuntoli, I., Robinson, E. L., Clark, D. B., Arnell, N. W., Dankers, R., ... & Wisser, D. (2014). Hydrological droughts in the 21st century, hotspots and uncertainties from a global multimodel ensemble experiment. *Proceedings of the National Academy of Sciences*, *111*(9), 3262-3267.

Salas, J. D., & Obeysekera, J. (2014). Revisiting the concepts of return period and risk for nonstationary hydrologic extreme events. *Journal of hydrologic engineering*, *19*(3), 554-568.

Satoh, Y., Yoshimura, K., Pokhrel, Y., Kim, H., Shiogama, H., Yokohata, T., ... & Oki, T. (2022). The timing of unprecedented hydrological drought under climate change. *Nature communications*, *13*(1), 3287.

Van Loon, A. F., & Van Lanen, H. A. (2012). A process-based typology of hydrological drought. *Hydrology and Earth System Sciences*, *16*(7), 1915-1946.

---

## Author Comment (AC3)

**Reply to Comments by Referee #3**

**Review of "Evolution of nonstationary hydrological drought characteristics in the UK under warming" by Jha et al.**

**Comment** This manuscript proposes an analysis of the probability of occurrence of rare hydrological droughts in the UK under different warming levels. It makes use of state-of-the-art multimodel hydrological projections over the 21st century and of an extreme value analysis of drought characteristics (duration, intensity, severity). Note that hydrological drought are defined through daily anomalies with respect to an average daily regime and the manuscript topic is therefore not about low-flows. The manuscript is well written and well organised. Methods, results, and corresponding conclusions are sound. Moreover, such an analysis is timely when climate change adaptation is more and more organised around Global or Regional Warming Levels (see e.g. Sauquet et al., 2025). **I have however one concern about the seasonal approach chosen to present results**. It is detailed -- with suggestions of improvements -- along another general comment below. Specific comments are also detailed afterwards. I would therefore recommend a revision to be made on this point before publication.

**Reply:** We sincerely thank the reviewer for their thorough reading and detailed evaluation of our manuscript. We greatly appreciate your positive recommendations, questions, and suggesting constructive changes for improvement. Below, we provide point-wise responses to each of your comments.

**General comments**

**Comment 1: L244-249: This is where the methodological choices seem rather strange.** I do not understand why results are artificially broken by seasons after an event-based analysis. Indeed, any physical continuity across seasons is lost while this continuity is of uttermost importance when analysing seasonal shifts in drought development. Furthermore, analysing seasons separately leads inevitably to an artificial upper bound for duration (and therefore severity). **I would strongly suggest considering instead e.g. the proportion of each individual event in each season (possibly considering also different consecutive years/seasons for a single multiyear event).** This would only slightly alter the analysis and bring much more insight into drought changes.

**Reply 1:** Thank you for raising this very important point. We would like to clarify that in our study, event detection is performed on the full continuous time series in reference period and warming level periods, not within seasons. Seasonal metrics are calculated only after drought events and their onset are identified, so physical continuity is preserved, and duration or severity are not artificially capped by seasonal or yearly boundaries except in the last year of the period. We have calculated the drought characteristics based on the starting point of the event and assigned the season based on starting month. While considering the proportion of each event across seasons could add another dimension and provide additional insights, it might also complicate both the representation and interpretation of drought events, especially for some long or severe events that span multiple seasons. This might also require different approaches for allocating proportionality to duration, severity, and intensity, potentially leading to multiple representations of the same event. For example, a 150-day event might need to be expressed as 70% in one season, 20% in another, and 10% in a third based on its duration distribution. However, the same event could show a different distribution in terms of severity or deficit volume such as peaking in the final season which would then need to be accounted for. In some cases, this potentially dilute the key message that the event is after all fundamentally only one long and continuous event.

For these reasons, we assign the season based on the starting month of each drought event. Additionally, the use of a 30-day rolling-window and variable threshold methods allows us to incorporate smooth intra-annual variability and ensures that hydrological droughts are identified only when flows fall below the seasonally expected level for that day.

In the revised manuscript, we aim to clarify our seasonal approach more clearly by mentioning the above points in **L244-249** and also include an additional analysis comparing the distribution of drought characteristics using alternative metrics as suggested alongside the approach we adopted to assess the suitability and sensitivity of the two different methods.

**Comment 2:** The warming level analysis is rather attractive, but **associated uncertainties are barely discussed** in the manuscript although they are crucial for adaptation purposes. Some specific comments below detail a few aspects of that, but the main question arising here is the relative importance of **the diversity of climate projections, the diversity of hydrological models, and GEV parameter estimation**. I may have missed corresponding explanations, but it is unclear **how pooling is done across climate projections** (and or not across hydrological models) to derive GEV estimates. **Making hypotheses even clearer** is a major point of potential improvement for the manuscript.

**Reply 2:** Thank you for the important observation and suggestions. We agree that associated uncertainty is one of the important aspects on our analysis and needs more discussion in the manuscript. In the revised version, we intend to include points to explain the methodological basis of addressing this and more discussion based on the results we obtained. More specifically:

on the importance of the 'diversity of climate projections'-

we will modify **L131-135** in Section 2.1 explaining eFLaG 'simrcm' projections data as following:

The 'simrcm' projections consist of a 12-member ensemble generated using perturbed-parameter runs of the Hadley Centre global climate model (GCM, HadGEM3-GC3.05) and regional climate model (RCM, HadREM3-GA705) (Murphy et al., 2018). Each ensemble member represents a plausible variation in model parameters to capture uncertainty in the climate response, while all members share the same underlying model framework and follow the high-emissions scenario (RCP8.5). We focus exclusively on the RCP8.5 pathway in this study because it represents a worst-case, but varying warming trajectories, allowing a clear assessment of the upper bound of potential changes in extremes under climate change.

on the importance of the 'diversity of hydrological models'-

In the manuscript, we provide a brief description of the structure and characteristics of the hydrological models used (**L136-144**). In our analysis, we observe that the uncertainty in return-level changes is driven primarily by the rarity and the characteristic of the hydrological drought events rather than by differences between the individual hydrological models. We illustrate this in Figure 6 (main text) and Figure S4a-f (supplementary material), where we compare stationary and nonstationary return-level changes across UK. Since these figures show that inter-model differences are relatively smaller as compared to the factors mentioned above, we chose not to discuss the results of each hydrological model separately and clarified this point in **L406-408**.

However, in the revised manuscript, we will introduce a more detailed analysis of uncertainty by including additional results. We will incorporate and discuss a new figure in the main text (shown below) illustrating changes across models, seasons, characteristics, warming levels, and return periods, providing a comprehensive perspective on the associated uncertainties and explicitly highlighting inter-model differences for nonstationary return levels. This will help readers better understand the

range of possible deviations and uncertainties in hydrological drought projections across different layers, with particular emphasis on variations between models.

[Figure]

On the importance of 'GEV parameter estimation'-

Parameter uncertainty is a key aspect of the nonstationary hydrological drought risk assessment. As discussed in the manuscript (Figure S1), we obtain robust posterior distributions of parameters using a Bayesian MCMC approach. To illustrate the impact of parameter uncertainty, we computed results using four different summaries of the parameter distribution: the 25th percentile (Q25), the 75th percentile (Q75), the mean, and the median. The estimates of return level changes, as well as the differences between nonstationary and stationary return levels across these four summaries, demonstrate consistency and robustness throughout the analysis, as shown in Figures S2 and S3. In the revised manuscript, we will provide a more detailed description of these points and will also discuss the possible implications of the same.

In addition to explaining the three factors individually, we will also discuss their relative importance in the revised manuscript.

On pooling procedure:

In the revised manuscript, we will modify **L230-238** as follows to provide a clearer and more precise explanation:

For each of the 12 ensemble members of each hydrological model, we first calculated the daily mean flow values for every day of the reference period using the eFLaG dataset. We then applied a 30-day rolling window centred on each day of the year. For example, for 15 January, the window includes flows from 15 days before to 15 days after. This smoothing method helps capture natural variability in daily

flows and prevents the resulting statistics from being overly influenced by short-lived extreme events. Using these rolling-window values, we derived 365 Q90 thresholds, one for each day of the year, representing the 90th percentile exceedance flow for the reference period. These thresholds were then used as the baseline against which projected flow levels at different warming levels were compared. Specifically, we calculated the difference between projected flows and the corresponding daily Q90 threshold to identify high-flow anomalies or deficits relevant for drought analysis. The resulting drought characteristics for each warming level were subsequently pooled across all 12 ensemble members, and this pooled dataset was used to fit GEV distributions to assess changes in extremes under future climate conditions.

On Making hypotheses clearer:

As also requested by Reviewer 2, we propose to modify **L186** and include the following lines in the revised manuscript to provide additional clarification:

The null hypothesis in our study assumes that drought characteristics extremes are stationary, meaning their statistical properties do not change over time or with temperature. Using the likelihood ratio test, this hypothesis is evaluated by comparing the fit of stationary and nonstationary GEV models. The null hypothesis is rejected when the p-value falls below 0.05, indicating that including temperature as a covariate significantly improves the model. Such an approach is consistent with standard methods in extreme value analysis for hydrological data (Salas and Obeysekera, 2014; Das and Umamahesh, 2017).

**Specific comments**

**Comment 3:** L146-150: Does eFlag use CHESS-SCAPE as forcings for hyrological models? Please make it clear.

**Reply 4: L146-150** will be modified as follows:

The CHESS-SCAPE temperature records used in this analysis are derived from UKCP18 projections that have been downscaled to 1 km resolution using methods that account for local topographic effects and pattern scaling properties for different scenarios. It should be noted that the eFLaG dataset is based directly on the original UKCP18 projections.

**Comment 4:** L159-161: How is the daily temperature anomaly defined in CHESS-SCAPE? What is the reference period? How is seasonality taken into account? What is the spatial scale considered for computing the anomalies: local, UK, global? Please detail the answers explicitly in the manuscript.

**Reply 5:** Thank you for the suggestion. More clarification at **L161** will be added in the revised version of the manuscript as below:

Here, daily temperature anomaly for each period were calculated relative to the mean temperature over the UK for the reference period (1989-2018). After identifying drought events, we matched the timestamp of each drought characteristic with the corresponding temperature time series and used the mean reference-period to compute the anomalies, which were then used as covariates. Please refer to Section 2.4 for further details on the event-calculation methodology to understand how seasonality and continuation of events have been considered.

**Comment 5:** L 220-224: Please define exactly what these warming levels refer to. Are they Global Warming Levels (GWLs)? Are they Regional (UK) Warming Levels (RWLs)? In the first case, how GWLs translate into RWLs?

**Reply 5:** We agree that clearer explanation of the considered warming levels is needed. In addition to replacing the word 'global' with 'regional warming levels/UK warming levels', we propose to add the following statements with suitable references at **L220-224** in the revised manuscript clarifying warming level's relationship to the underlying global temperature used during CHESS-SCAPE dataset:

The warming levels in this analysis should be interpreted as regional UK warming levels rather than global warming levels, since CHESS-SCAPE provides only UKCP18 climate projections over the UK. While the CHESS-SCAPE framework does use global mean air temperature from UKCP18 GCMs and uses time shifting and pattern scaling, the downscaled dataset contains only UK specific surface variables (Robinson et al., 2022a). However, these warming levels are broadly aligned with GWLs as UKCP18 assumes seasonal UK climate anomalies scale linearly with global mean temperature, and it is known that UK temperature changes generally track global land-surface warming (Kendon et al., 2023).

**Comment 6:** L238-251: How does this pooling procedure compare with the Sequent Peak Algorithm (SPA) traditionally used with a fixed threshold?

**Reply 6:** The pooling methodology used in our analysis differs from the traditional Sequent Peak Algorithm (SPA). In SPA, a fixed threshold is applied, and a new drought event is recognized only when the cumulative deficit returns to zero. In contrast, our approach allows two drought events separated by a single day to be pooled if the flow on that day does not exceed the cumulative deficit accumulated beforehand. This ensures that longer droughts with short interruptions are properly captured, which might otherwise be overlooked by the SPA-based method.

**Comment 7:** L242: What does "standard" refer to? 30 days sound already quite long for a drought event, even with a daily varying threshold. Please comment on that. Plus, this introduces a hard lower bound for duration, which may hide some signal on "flash droughts". I would therefore recommend not censuring a priori such short events which might (depending on catchment storage) bring in some extreme values of e.g. intensity.

**Reply 7:** Thank you for the comment and recommendation. In the revised manuscript, we will clarify this and discuss its implications, including the potential exclusion of flash droughts. We propose to include the following clarification at **L242:**

To reduce uncertainty arising from very short, potentially non-significant drought events caused by daily variability in the threshold, we excluded events with a duration of less than 30 days. Given that we focus on Q90 to derive these events, even after applying precautionary measures such as a 30-day moving window and a 12-member ensemble pool to ensure smoother and larger sample sizes, extreme value analysis remains challenging, particularly for rare, small drought events. We acknowledge that this threshold effectively imposes a hard lower bound on drought duration and may also exclude smaller events such as flash droughts. Nevertheless, we chose 30 days which has widely been used in similar analyses (Anderson et al., 2025; Brunner and Chartier-Rescan, 2024), as compromise to balance robustness of event statistics with capturing meaningful hydrological droughts.

**Comment 8:** Section 3.1. and Fig. 2.: Fig. 2 presents strong fluctuations of stationarity properties across warming levels with e.g. the number of nonstationary catchments declining from 1.5°C to 2°C and increasing to 3°C. This demonstrates the limits of pattern scaling properties and what is striking is that this is barely commented in the corresponding text of Sect. 3.1. Please at least add such comments on that, as pattern scaling is indeed one of the foundations of this manuscript. What is also missing here is the corresponding spatial patterns of such nonstationarity, which would be interesting for understanding the limits of pattern scaling across the UK.

**Reply 8:** We agree that limitations of pattern scaling properties and spatial patterns of nonstationary should be discussed in more detail in Section 3.1. In the revised manuscript, we will modify this section and **L266-270** as following for more clarity and explanation:

Further, the fluctuations in the stationarity properties of catchments specifically, the number of nonstationary catchments declining from 1.5°C to 2°C warming but then increasing at 3°C highlight the limitations of the pattern scaling assumption. This is central to CHESS-SCAPE and UKCP18 data considered, which is based on the assumption that local or regional climate responses scale linearly with global mean temperature (Robinson et al., 2022a). The observed variations suggest that this assumption may break down for certain warming levels or in specific regions, as illustrated in Figures 2 and 3. Examining the spatial distribution of nonstationarity across the UK provides insight into where pattern scaling might hold and where caution is needed, highlighting regions dominated by nonlinear responses. Therefore, changes in nonstationary properties, their dependence on warming levels, catchment characteristics, and seasonal variability must be considered with full caution when modelling the evolution of future hydrological droughts.

**Comment 9: L.288**: Fig. S1 is rather required in the main text in my view.

**Reply 9: Figure S1** will be included in the main text in the revised manuscript.

**Comment 10:** Fig. 3: This is clearly unreadable as such, for several reasons. One, the color scale is non linear, which is definitely against perceptual rules (Hawkins, 2015 ; Stoelze and Stein, 2021). Two, maps are way too small. Three, using catchment surface as the support for colors is not appropriate for such small figure dimensions, as it perceptually highlights only large catchments. I would therefore strongly suggest using shapes at the outlet of each catchment. Four, colorscale label sizes are not homogeneous across facets, and the colorscale titles "location paramater" are redundant.

**Reply 10:** Thank you for the constructive feedback. Here we have included the modified figure based on your recommendations for your reference. If suggested, we will replace Figure 3 in the revised manuscript with this new figure.

[Figure]

**Figure 3.** Mean and standard deviation of location parameter samples for GR4J model during summer season at 3°C warming level.

**Comment 11:** Fig. 4: This is again much too small for being readable. And again, I would strongly recommend using e.g. disc shapes, and also using grey instead of black for UK coastlines. A discrete color scale might also be more effective.

**Reply 10:** Thank you for your recommendation. We understand that it may be challenging for readers to interpret the distribution of changes in return levels for smaller catchments in the current form of plot. Main aim of our analysis is to provide an overall picture of changes in different drought characteristics, seasons, warming levels, and return periods across the UK. Each figure combination therefore contains multiple smaller plots of the UK, resulting from iterative analyses providing readers an opportunity to compare multiple cases. However, we do understand that this can be improved based on your suggestions. Below we have included a sample modified Fig. 4 incorporating the recommendations about using discs instead of colouring the catchment shapefiles, a discrete color scale and grey for UK coastlines in this document. We also propose to split Fig. 4a and Fig 4b into two separate and larger figures as given below. This, if preferred, may replace Fig. 4 in the revised manuscript, and similar updates mat be applied to the other figures in the supplementary information:

[Figure]

**Figure 4a.** Percentage change in mean nonstationary 10-year return levels for different drought characteristics across all warming levels and seasons.

[Figure]

**Figure 4b.** Percentage change in mean nonstationary 500-year return levels for different drought characteristics across all warming levels and seasons.

**Comment 12:** Fig. 5: Please confirm (in the legend) that boxplots show differences across catchments only (and not across climate/hydrological models).

**Reply 12:** Thank you for the suggestion. We will further clarify that the boxplot represent difference across catchments in the revised manuscript.

**Comment 13:** L371-372: What can the reader refer to when reading that rarer droughts are accompanied by large variability? Across what? Spatial? Parameter estimation? Other?

**Reply 12:** These lines describe Figure 5a,b, showing that the variability in percentage change of return level estimates increases with increasing return period. To add more explanation, we propose to modify line **L371-L372** as follows:

It can also be concluded from Figure 5a,b that rarer droughts, which are inherently associated with larger uncertainty contributed by factors such as event identification, estimation of distribution parameters, or an interaction of these factors, are not only associated with larger changes but also with greater overall spatial variability across catchments.

**Comment 14:** L421: Please provide factual elements to comfort the assertion of "robust estimates of uncertainty".

**Reply 14:** Thank you for the comment, in the revised version of the manuscript, we will modify the **L418-423** as following:

Despite this, the findings from this analysis give crucial insights about the changing future hydrological drought characteristics in the UK under climate change. The results not only quantify the changes in the return level of drought duration, severity, and intensity but also provide explicit estimates of uncertainty in the GEV distribution parameters and associated return levels centred on the methodological framework adopted in this study. The Bayesian approach allows full posterior distribution of the GEV parameters to be explored, enabling return level estimates to be assessed across a wide range of parameter values. This is further supported by using MCMC simulations whose convergence is diagnosed with the Heidelberger-Welch test, which helps to ensure that the posterior distributions are stable and reliable. These elements along with moving window approach and pooling procedure to identify drought events ensure that thorough attention has been given from the initial drought identification through to the estimation of return levels, resulting in reliable and transparently quantified estimates of return level across temporal scales, models, seasons and warming levels.

**References**

Hawkins, E. (2015) Graphics: Scrap rainbow colour scales. Nature, 519, 291. https://doi.org/10.1038/519291d

Sauquet et al. (2025), Évolution de l'hydrologie de surface en France par niveau de réchauffement, https://doi.org/10.57745/MN29RG, Recherche Data Gouv, V5.

Stoelzle, M. & Stein, L. (2021) Rainbow color map distorts and misleads research in hydrology -- guidance for better visualizations and science communication. Hydrology and Earth System Sciences, 25, 4549-4565. https://doi.org/10.5194/hess-25-4549-2021

**REFERENCES:**

Anderson, B. J., Muñoz-Castro, E., Tallaksen, L. M., Matano, A., Götte, J., Armitage, R., ... & Brunner, M. I. (2025). What is a drought-to-flood transition? Pitfalls and recommendations for defining consecutive hydrological extreme events. Hydrology and Earth System Sciences, 29(21), 6069-6092.

Brunner, M. I., & Chartier-Rescan, C. (2024). Drought spatial extent and dependence increase during drought propagation from the atmosphere to the hydrosphere. Geophysical Research Letters, 51(6), e2023GL107918.

Das, J., & Umamahesh, N. V. (2017). Uncertainty and nonstationarity in streamflow extremes under climate change scenarios over a river basin. Journal of Hydrologic Engineering, 22(10), 04017042

Kendon, M., Doherty, A., Hollis, D., Carlisle, E., Packman, S., McCarthy, M., ... & Sparks, T. (2024). State of the UK Climate 2023. International Journal of Climatology, 44, 1-117.

Murphy, J. M., Harris, G. R., Sexton, D. M. H., Kendon, E. J., Bett, P. E., Clark, R. T., Eagle, K. E., Fosser, G., Fung, F., and Lowe, J. A.: UKCP18 land projections: science report, 2018.

Robinson, E. L., Huntingford, C., Shamsudheen, S., and Bullock, J.: CHESS-SCAPE: Future projections of meteorological variables at 1 km resolution for the United Kingdom 1980-2080 derived from UK Climate Projections 2018, NERC EDS Centre for Environmental Data Analysis, https://doi.org/http://dx.doi.org/10.5285/8194b416cbee482b89e0dfbe17c5786c, 2022

Salas, J. D., & Obeysekera, J. (2014). Revisiting the concepts of return period and risk for nonstationary hydrologic extreme events. Journal of hydrologic engineering, 19(3), 554-568.

.